# HL156A, an AMP-Activated Protein Kinase Activator, Inhibits Cyst Growth in Autosomal Dominant Polycystic Kidney Disease

**DOI:** 10.3390/biom14070806

**Published:** 2024-07-07

**Authors:** Sujung Seo, Hyunho Kim, Jung-Taek Hwang, Jin Eop Kim, Jisu Kim, Sohyun Jeon, Young-jin Song, Kwang-ho Choi, Gwangeon Sim, Myunkyu Cho, Jong-woo Yoon, Hyunsuk Kim

**Affiliations:** 1Department of Internal Medicine, Hallym University Medical Center, Chuncheon Sacred Heart Hospital, Chuncheon 24253, Republic of Korea; asasa4444@naver.com (S.S.); drakehjt@hallym.or.kr (J.-T.H.); enonakajo@hallym.or.kr (J.E.K.); 230073@hallym.or.kr (J.K.); so721@hallym.or.kr (S.J.); songyj141@hallym.or.kr (Y.-j.S.); sindowhikaru@hallym.or.kr (K.-h.C.); 230162@hallym.or.kr (G.S.); whausrb@hallym.or.kr (M.C.); yoonjw@hallym.ac.kr (J.-w.Y.); 2Center for Medical Innovation, Biomedical Research Institute, Seoul National University Hospital, Seoul 03080, Republic of Korea; 83150@snuh.org

**Keywords:** ADPKD, AMPK activator, HL156A, collecting duct-specific *Pkd1* KO mice

## Abstract

Background: Autosomal dominant polycystic kidney disease (ADPKD) is the most prevalent genetic kidney disorder. While metformin has demonstrated the ability to inhibit cyst growth in animal models of ADPKD via activation of adenosine monophosphate-activated protein kinase (AMPK), its effectiveness in humans is limited due to its low potency. This study explored the impact of HL156A, a new and more potent AMPK activator, in a mouse model of ADPKD. Methods: To investigate whether HL156A inhibits the proliferation of renal cyst cells in ADPKD in vitro, exogenous human telomerase reverse transcriptase (hTERT)-immortalized renal cyst cells from ADPKD patients were treated with HL156A, and an MTT (dimethylthiazol-diphenyltetrazolium bromide) assay was performed. To assess the cyst-inhibitory effect of HL156A in vivo, we generated *Pkd1* conditional knockout (KO) mice with aquaporin 2 (AQP2)-Cre, which selectively expresses Cre recombinase in the collecting duct. The effectiveness of HL156A in inhibiting cyst growth and improving renal function was confirmed by measuring the number of cysts and blood urea nitrogen (BUN) levels in the collecting duct-specific *Pkd1* KO mice. Results: When cyst cells were treated with up to 20 µM of metformin or HL156A, HL156A reduced cell viability by 25% starting at a concentration of 5 µM, whereas metformin showed no effect. When AQP2-Cre male mice were crossed with *Pkd1*^flox/flox^ female mice, and when AQP2-Cre female mice were crossed with *Pkd1*^flox/flox^ male mice, the number of litters produced by both groups was comparable. In collecting duct-specific *Pkd1* KO mice, HL156A was found to inhibit cyst growth, reducing both the number and size of cysts. Furthermore, it was confirmed that kidney function improved as HL156A treatment led to a reduction in elevated BUN levels. Lastly, it was observed that the increase in AMPK phosphorylation induced by HL156A decreased ERK phosphorylation and α-SMA expression. Conclusion: HL156A has potential as a drug that can restore kidney function in ADPKD patients by inhibiting cyst growth.

## 1. Introduction

Autosomal dominant polycystic kidney disease (ADPKD) is the most prevalent genetic kidney disease, affecting from 1 in 500 to 1 in 1000 individuals. It is a severe condition, with 50% of patients over the age of 60 requiring renal replacement therapy [1,2]. In ADPKD, numerous fluid-filled cysts form in the kidneys, compressing the renal parenchyma and leading to progressive renal failure [3]. In addition, ADPKD is a systemic disease that can cause a variety of organ-related issues, including cardiovascular disease and cysts in the liver and pancreas [3,4]. The disease is caused by mutations in the *PKD1* and *PKD2* genes, which encode polycystin-1 (PC1) and 2 (PC2) proteins; besides *PKD1* and *PKD2*, other genes are associated with PKD [5]. Mutations in *PKD1* and *PKD2* account for 85% and 15% of ADPKD patients, respectively [6,7]. The phenotype associated with *PKD1* mutations is known to be more severe than that associated with *PKD2* mutations [6].

The study of ADPKD has advanced significantly since the discovery of the *PKD1* and *PKD2* genes. Several signaling pathways related to cyst growth have been identified, leading to the development of drugs that target these pathways. There is considerable potential for therapeutic development and clinical translation by focusing on these key signaling pathways [8]. Increased cAMP signaling has been identified as a key mechanism for cystic growth and fluid secretion. This understanding has led to the development of treatments based on vasopressin V2 receptor (V2R) antagonism. Tolvaptan, the primary therapeutic drug in this category, works by reducing cAMP levels, which slows cyst growth and delays the loss of renal function [9]. However, tolvaptan is an expensive medication and is associated with several side effects, including polyuria, fatigue, and reversible transaminase elevations. Consequently, its use is generally restricted to high-risk ADPKD patients [10,11].

Adenosine monophosphate-activated protein kinase (AMPK) is a crucial cellular sensor that regulates a variety of cellular processes, including cellular energy and metabolic homeostasis. AMPK is a heterotrimeric enzyme complex consisting of α, β, and δ subunits, and it is activated through the phosphorylation of Thr172 on the α subunit. The activation of AMPK plays a role in cytokine production, the regulation of inflammation, cell proliferation, and cell death [12].

Metformin, an antidiabetic drug, indirectly activates AMPK by inhibiting mitochondrial function and reducing cellular adenylate charge through its interaction with the AMPK gamma isoform. It also inhibits tumor growth by inhibiting mTOR activity and has demonstrated anti-fibrotic effects in animal models of liver fibrosis [13,14]. Studies on the effects of metformin in ADPKD have shown a reduction in cystic disease in a tamoxifen-inducible *PKD1*^del^ conditional KO model administered high-dose metformin (300 mg/kg) [15]. In clinical trials, metformin slowed the rate of glomerular filtration rate (GFR) decline in adults with ADPKD, although the results were not significant [16,17]. However, despite its numerous beneficial properties, metformin’s hydrophilic nature limits its cellular penetration, posing challenges for its use as an anti-cancer drug [13].

HL156A is a novel derivative of metformin designed with increased hydrophobicity to address the limitations of metformin, enabling it to induce AMPK phosphorylation more effectively than either metformin or AICAR (5-aminoimidazole-r-carboxamide ribonucleotide) [18,19,20]. HL156A has been reported to have protective effects against peritoneal, liver, and renal fibrosis [18,19,21]. Additionally, HL156A has demonstrated potential effects against MYC-dependent lymphoma [22] and glioblastoma [23]. Given its superior AMPK activation compared to metformin and its lower side effects despite its higher potency, HL156A is a promising candidate drug for treating human ADPKD. The aim of this study was to evaluate the efficacy of HL156A, a potent AMPK activator, in a mouse model of ADPKD.

## 2. Materials and Methods

### 2.1. MTT Assay in Renal Cyst Cells Treated with HL156A

To confirm the viability of human telomerase reverse transcriptase (hTERT)-immortalized renal cyst cells [24] by treatment with HL156A, an MTT (dimethylthiazol-diphenyltetrazolium bromide) assay was performed (Sigma, St. Louis, MO, USA). Cells were seeded into a 96-well plate in 100 µL of growth media at 0.5 × 10^5^ cells/well for 24 h. Cells were then incubated with a range of HL156A concentrations—from 0 µM to 20 µM—for 24 h. Then, after removing the medium from each well, 100 µL of MTT solution (mg/mL in serum-free media) was added to each well and incubated for 2 h at 37 °C. After removing the MTT solution, 100 µL of DMSO (dimethyl sulfoxide) was added to each well to dissolve the formazan crystal and measure the absorbance at 540 nm by using a GloMax^®^ Discover multimode plate reader (Promega, Madison, WI, USA).

### 2.2. Generation of Collecting Duct-Specific Pkd1 KO Mice

*Pkd1*^flox/flox^ mice (B6.129S4-*Pkd1^tm2Ggg^*/J) and AQP2-Cre mice (B6.Cg-Tg(Aqp2-cre)1Dek/J) were purchased from Jackson Laboratories. The collecting duct-specific *Pkd1* KO mice were generated in a manner similar to that previously described [25,26]. Floxed *Pkd1* mice have loxP sites on exons 1 and 4 of the *Pkd1* gene, which allow for the selective deletion of essential coding regions when exposed to Cre recombinase. AQP2-Cre mice were used to create a *Pkd1* KO model specific for the collecting duct. First, AQP2-Cre mice were mated with floxed *Pkd1* mice, and offspring that were positive for AQP2-Cre and heterozygous for *Pkd1* (*Pkd1*^flox/+^) were selected. These mice were mated with homozygous floxed *Pkd1* (*Pkd1*^flox/flox^) to obtain an ADPKD model in *Pkd1* KO mice [25]. This animal study was conducted according to the guidelines of the Institutional Animal Care and Use Committees (IACUC) of Hallym University (HallymR12021-32) after receiving approval.

### 2.3. Genotyping

Genomic DNA was extracted from mouse tails using the DNeasy Blood & Tissue Kit (Qiagen, Hilden, Germany). The samples were then stored at −80 °C until further analysis. Polymerase chain reaction was performed on the tail genomic DNA using the following primers: AQP2-Cre forward: 5′-CTC TGC AGG AAC TGG TGC TGG-3′; AQP2-Cre reverse: 5′-GCG AAC ATC TTC AGG TTC TGC GG-3′; Pkd1 forward: 5′-CCT GCC TTG CTC TAC TTT CC-3′; Pkd1 reverse: 5′-AGG GCT TTT CTT GCT GGT CT-3′ [27,28].

### 2.4. Administration of HL156A

The mice were divided into a control group (AQP2-Cre;*Pkd1*^flox/+^), which did not develop cysts in the kidneys, and a *Pkd1* KO group (AQP2-Cre;*Pkd1*^flox/flox^), which developed cysts in the kidneys. When mice were injected with a dose of 30 mg/kg of HL156A starting from P2, they did not survive, so the experiment was conducted with a dose of 25 mg/kg or less. HL156A (15 mg/kg or 25 mg/kg) was administered orally to mice every other day from P2 to P28. Distilled water was used as the vehicle.

### 2.5. Western Blot

The kidney tissues of mice were homogenized using a lysis buffer that included complete protease inhibitor (Roche, Basel, Switzerland) and phosphatase inhibitors (Sigma). The homogenates were then centrifuged at 12,000× *g* for 20 min at 4 °C, and the supernatants were collected for further use. Protein concentrations in the lysates were determined using the Bradford method. The protein samples were subjected to sodium dodecyl sulfate–polyacrylamide gel electrophoresis (SDS-PAGE) and subsequently transferred onto polyvinylidene difluoride (PVDF) membranes. These membranes were incubated with primary antibodies, including anti-AMPK, anti-phospho-AMPK, anti-smooth muscle actin (SMA), anti-extracellular signal-regulated kinase (ERK), and anti-phospho-ERK (Cell Signaling, Danvers, MA, USA). Horseradish peroxidase (HRP)-conjugated secondary antibodies (ACE Biolabs, Jacksonville, FL, USA) were then applied, and the proteins were visualized using chemiluminescence (Amersham Biosciences, Amersham, UK).

### 2.6. Histology and Immunofluorescence

The mice were sacrificed at P28. The kidney tissues were fixed in 4% paraformaldehyde, and any remaining paraformaldehyde in the tissue was removed through a series of hydration and dehydration steps. The kidney tissue block was then created by embedding it in paraffin. Sections of the kidney tissue, 5 µm thick, were cut from the paraffin-embedded blocks and mounted on coated slides. Subsequently, hematoxylin and eosin (H&E, Baton Rouge, LA, USA) staining was performed to assess the number and size of cysts, following the manufacturer’s instructions.

To confirm whether cysts were generated only in the collecting duct of the *Pkd1* KO mice, fixed kidney tissue slides were permeabilized with 0.1% Triton X-100 in PBS for 10 min, followed by blocking with 1% bovine serum albumin (BSA) in phosphate-buffered saline (PBS) for 30 min. Thereafter, the kidney tissue slides were incubated with primary antibodies in a blocking buffer (5% BSA/0.05% Triton X-100 in PBS) overnight. We used FITC-conjugated anti-lotus tetragonolobus lectin (LTL, Vector Laboratories, Newark, CA, USA) as a proximal tubule marker and rhodamine-conjugated anti-dolichos biflorus agglutinin (DBA, Vector Laboratories) as a collecting duct marker.

### 2.7. Analysis of Blood Urea Nitrogen

For the analysis of blood urea nitrogen (BUN), blood samples were collected from the intraorbital vein using a microcapillary tube at the time of sacrifice. These samples were centrifuged at 1500× *g* for 15 min within 1 h of collection to separate the plasma. The plasma samples were then stored at −80 °C until further analysis. BUN levels were determined using a BUN Colorimetric Detection Kit (Thermo Fisher Scientific, Waltham, MA, USA), according to the manufacturer’s instructions. The absorbance of each sample was measured at 450 nm using a GloMax^®^ Discover multimode plate reader (Promega).

### 2.8. Statistics

All statistical analyses were conducted using IBM SPSS Statistics ver.25 (IBM Corp., Armonk, NY, USA). One-way analysis, the Mann–Whitney test, and the Jonckheere–Terpstra test were used to compare values as appropriate.

## 3. Results

### 3.1. HL156A Inhibits ADPKD Renal Cyst Cell Proliferation More Effectively Than Metformin

An MTT assay was conducted to determine if HL156A, a novel AMPK activator, can inhibit the proliferation of renal cyst cells effectively in smaller quantities compared to metformin, which is typically used in larger amounts for ADPKD. hTERT-immortalized renal cyst cells were exposed to varying concentrations of metformin or HL156A (Figure 1). While metformin did not decrease cell viability, HL156A reduced it by approximately 25% at a concentration of 5 µM. Thus, HL156A is capable of inhibiting cyst cell proliferation at significantly lower concentrations than metformin.

### 3.2. Characterization of the Collecting Duct-Specific Pkd1 KO Mice

Since mice with homozygous mutations of *Pkd1* die in utero, we generated an ADPKD mouse model in which *Pkd1* is conditionally knocked out only in the collecting duct by conjugating Cre recombinase to the promoter of the *AQP2* gene, which is expressed only in the collecting duct of the kidney [25]. We generated the collecting duct-specific *Pkd1* KO mice by crossing two types of B6 mice (AQP2-Cre and *Pkd1*^flox/flox^). Crossbreeding AQP2-Cre;*Pkd1*^flox/+^ mice with *Pkd1*^flox/flox^ mice has a 25% chance of producing an ADPKD mouse model (AQP2-Cre;*Pkd1*^flox/flox^) (Figure 2A). The collecting duct-specific *Pkd1* KO mice (AQP2-Cre;*Pkd1*^flox/flox^) developed an overt ADPKD phenotype with kidney enlargement due to postnatal cysts (Figure 2B).

H&E staining was performed to evaluate cyst formation in each period. The staining revealed an increase in both the number and size of cysts in the kidney over time (Figure 2C). Immunofluorescence staining of kidney tissues confirmed that cyst formation was exclusively in the collecting duct (Figure 2D). To assess the fertility of AQP2-Cre mice by sex, 20 pairs of mice were used in the following configurations: AQP2-Cre;*Pkd1*^flox/+^ female and *Pkd1*^flox/flox^ male or AQP2-Cre;*Pkd1*^flox/+^ male and *Pkd1*^flox/flox^ female. When a *Pkd1* female and an AQP2-Cre male or a *Pkd1* male and an AQP2-Cre female were crossed, there was no difference in the number of litters between the two groups (Figure 2E).

### 3.3. HL156A Reduces the Kidney Weight-to-Body Weight Ratio in the Collecting Duct-Specific Pkd1 KO Mice

HL156A was administered orally on P2, before cyst formation, and then, the collecting duct-specific *Pkd1* KO mice were sacrificed on P28 to measure BUN levels (Figure 3A). In the collecting duct-specific *Pkd1* KO mice (AQP2-Cre;*Pkd1*^flox/flox^) that were administered 25 mg/kg HL156A from P2 to P28, the overall kidney size decreased (Figure 3B). Body weight (BW) and the ratio of kidney weight to body weight (KW/BW) are shown in Figure 3C. When normalized to body weight, there was no significant difference in the kidney weight in the control group (AQP2-Cre;*Pkd1*^flox/+^) according to the administration of HL156A. However, in the *Pkd1* KO group (AQP2-Cre;*Pkd1*^flox/flox^), the kidney weight decreased depending on the amount of HL156A. Therefore, HL156A reduced the kidney mass of the collecting duct-specific *Pkd1* KO mice.

### 3.4. HL156A Reduces the Number and Size of Cysts and Restores the Kidney Function of the Collecting Duct-Specific Pkd1 KO Mice

In the *Pkd1* KO group, the kidney weight of mice treated with HL156A decreased compared to the vehicle group. The number and size of cysts in the collecting duct-specific *Pkd1* KO mice significantly decreased after treatment with HL156A (Figure 4A,B). To confirm the effect of HL156A on improving renal function in the collecting duct-specific *Pkd1* KO mice, we analyzed BUN as a renal function-related marker. In the control group, there was no significant difference in BUN levels according to the administration of HL156A. In contrast, the *Pkd1* KO mice that received distilled water as the vehicle showed higher BUN levels than the control group. When HL156A was administered to the *Pkd1* KO group, the BUN levels decreased compared to the group that received the vehicle (Figure 4C). These results indicate that the kidney function of the collecting duct-specific *Pkd1* KO mice was restored.

### 3.5. Increased AMPK Phosphorylation by HL156A Suppresses the Phosphorylation of ERK and α-SMA Expression

To explore the signaling mechanisms involved in reducing the number and size of cysts and restoring kidney function, we investigated the role of ERK, which is associated with cyst proliferation, and α-SMA, which is linked to fibrosis. This was conducted through Western blot analysis using kidney lysates from collecting duct-specific *Pkd1* KO mice.

As the concentration of HL156A, a novel AMPK activator, increased, there was a corresponding increase in AMPK phosphorylation in the kidneys of collecting duct-specific *Pkd1* KO mice (Figure 5A). This finding suggests that oral administration of HL156A effectively induces AMPK phosphorylation in the kidneys of these mice.

We found that phosphorylation of ERK increased in the kidney of the collecting duct-specific *Pkd1* KO mice compared to control mice, but decreased after treatment with HL156A (Figure 5B). This result suggests that the reduction in the number and size of cysts after HL156A treatment may be related to the ERK signaling pathway. Additionally, the expression of α-SMA, which is related to fibrosis, increased in the collecting duct-specific *Pkd1* KO mice compared to control mice and then decreased after HL156A treatment (Figure 5C). Therefore, it can be inferred that HL156A suppressed fibrosis and restored kidney function in the collecting duct-specific *Pkd1* KO mice.

## 4. Discussion and Conclusions

In this study, we developed collecting duct-specific *Pkd1* knockout (KO) mice, which developed cysts exclusively in the collecting duct. We demonstrated that HL156A, a novel AMPK activator, not only inhibited cyst growth but also improved kidney function in these mice. In the collecting duct-specific *Pkd1* KO mice, treatment with HL156A led to a reduction in kidney mass, indicative of decreased cyst number and size, and restored kidney function, as evidenced by reduced BUN levels. Additionally, treatment with HL156A resulted in decreased phosphorylation of ERK, which is related to growth, and reduced expression of α-SMA, which is associated with fibrosis.

The role of AMPK in ADPKD has been extensively studied. In an ADPKD mouse model treated with 2-deoxyglucose, there was a notable reduction in kidney weight, volume, cystic index, and proliferation rate compared to untreated mice. This effect is attributed to the restoration of the ERK pathway by activated AMPK, which simultaneously inhibits the liver kinase B1 (LKB1)-AMP-AMPK axis and activates the mTOR complex 1 (mTORC1) [29]. Metformin significantly inhibited cystic growth in both in vitro and ex vivo models of renal cystogenesis and significantly reduced the cystic index in two mouse models of ADPKD [15]. In addition, studies have been conducted on salsalate [7], an ADaM site ligand, and caloric restriction [30]. However, no drugs are currently in clinical use, and the application of metformin in humans is challenging due to its minimal effect.

HL156A, which is a novel derivative of metformin, is a more powerful AMPK activator that complements the shortcomings of metformin [20]. A phase 1 study of HL156A in solid tumors has been completed and HL156A, also named IM156, demonstrated a good safety profile and tolerability at the recommended phase 2 clinical dose [31]. HL156A was shown to induce AMPK activation more strongly than metformin or AICAR [18,19]. HL156A reduced tubulointerstitial fibrosis in a unilateral ureteral obstruction model and inhibited the p-Smad3 downstream signaling cascade after TGF-β1 stimulation in vitro [19]. HL156A has the potential to be a candidate drug for ADPKD in humans due to its superior AMPK activation and fewer side effects compared to metformin.

In this study, we confirmed the effectiveness of HL156A in both in vitro and in vivo experiments for ADPKD. In vitro testing with hTERT-immortalized renal cyst cells demonstrated that HL156A more effectively inhibited cell viability than metformin. In vivo testing using collecting duct-specific *Pkd1* KO mice showed that HL156A reduced kidney mass by decreasing the number and size of renal cysts, thereby restoring kidney function. Additionally, an analysis of kidney protein expression levels related to cyst proliferation and fibrosis revealed that HL156A treatment resulted in decreased phosphorylation of ERK and reduced expression of α-SMA.

The ERK signaling pathway is known to be associated with cyst growth in ADPKD. It can be inferred that the significant increase in ERK phosphorylation observed in *Pkd1* KO mice, compared to control mice, contributes to cyst development in these mice, potentially related to the increase in kidney mass. It is reported that activated AMPK phosphorylates BRAF on serine 729 (S729). This phosphorylation disrupts their interaction with the scaffold protein Kinase Suppressor of RAS 1 (KSR1) and promotes the association of BRAF with 14-3-3 proteins, thereby attenuating the mitogen-activated protein kinase (MEK)/ERK signaling pathway [32]. Therefore, it is reasonable to conclude that activated AMPK by HL156A, an AMPK activator, phosphorylates BRAF and blunts the BRAF/MEK/ERK signaling pathway, and that the reduction in ERK phosphorylation by HL156A inhibits cyst growth and decreases kidney mass in *Pkd1* KO mice.

Renal fibrosis is known to impair kidney function, and our findings indicate that BUN levels in *Pkd1* KO mice were significantly higher than in the control group, suggesting diminished kidney function in these mice. Additionally, the increased expression of α-SMA, a marker associated with fibrosis, in *Pkd1* KO mice implies a link between fibrosis and elevated BUN levels. The observed reduction in α-SMA expression following treatment with HL156A suggests a decrease in fibrosis, potentially explaining the lowered BUN levels and indicating a restoration of kidney function in *Pkd1* KO mice. Therefore, we propose that HL156A may be a promising new drug alternative to metformin, which has not proven effective in clinical trials for ADPKD.

## Figures and Tables

**Figure 1 biomolecules-14-00806-f001:**
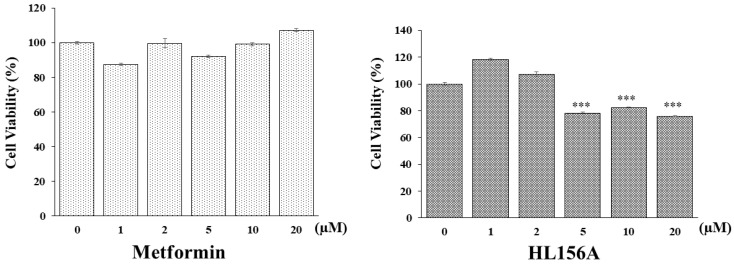
MTT assay for cell viability. Comparison of effects of metformin and HL156A on inhibition of cyst cell proliferation using exogenous human telomerase reverse transcriptase (hTERT)-immortalized renal cyst cells. HL156A inhibited cell proliferation by approximately 25% starting from 5 µM. *** *p* < 0.001.

**Figure 2 biomolecules-14-00806-f002:**
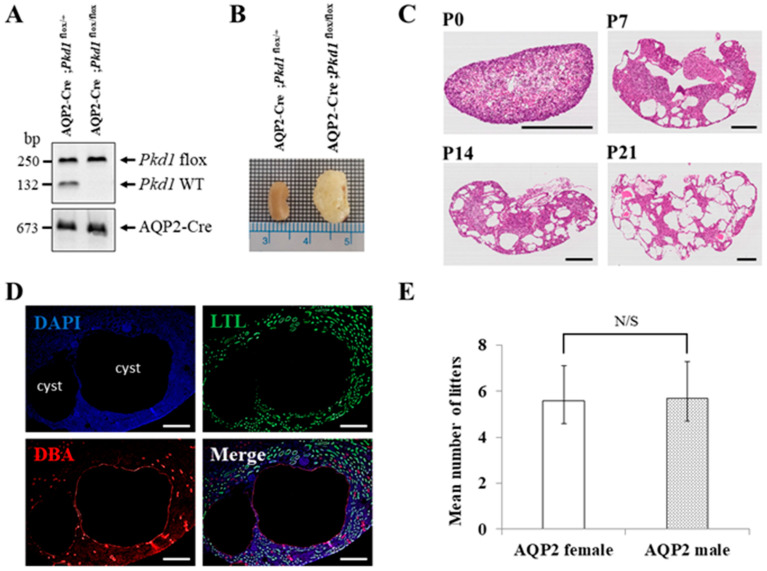
The characteristics of the collecting duct-specific *Pkd1* knockout (KO) mice. (**A**) Agarose gel analysis of PCR products amplified from genomic DNA of AQP2-Cre;*Pkd1*^flox/+^ and AQP2-Cre;*Pkd1*^flox/flox^ mice. Floxed *Pkd1*: 250 bp, *Pkd1* wild type (WT): 132 bp, and AQP2-Cre: 673 bp. (**B**) Gross morphology of the kidney. Scale bar, 1 mm. (**C**) Hematoxylin- and eosin-stained kidney tissues of AQP2-Cre;*Pkd1*^flox/flox^ mice for each period. (**D**) The origin of cysts. Immunofluorescence of kidney tissues of AQP2-Cre;*Pkd1*^flox/flox^ at P14 with LTL, a proximal tubule marker, and DBA, a collecting duct marker. Scale bar, 100 μm. (**E**) The fertility of female AQP2-Cre mice vs. male AQP2-Cre mice with *Pkd1*^flox/flox^ mice. N/S: not significant.

**Figure 3 biomolecules-14-00806-f003:**
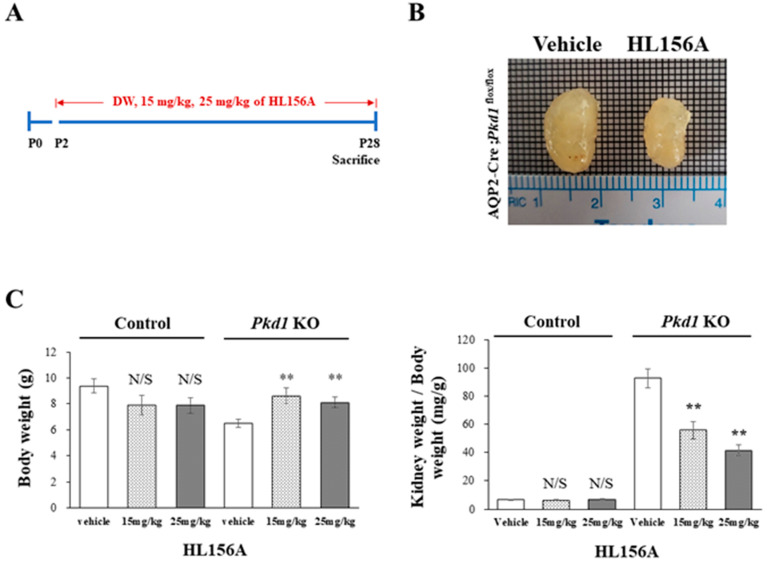
HL156A reduces the kidney mass of the collecting duct-specific *Pkd1* knockout (KO) mice. (**A**) A diagram of the HL156A schedule for the control (AQP2-Cre;*Pkd1*^flox/+^) and the *Pkd1* KO (AQP2-Cre;*Pkd1*^flox/flox^) mice. P: postnatal day. (**B**) Gross kidney morphology of P28 AQP2-Cre;*Pkd1*^flox/flox^ mice administered distilled water (DW) as the vehicle or 25 mg/kg of HL156A. (**C**) Body weight (BW) and the ratio of kidney weight to body weight (KW/BW) of the control (AQP2-Cre;*Pkd1*^flox/+^) and *Pkd1* KO (AQP2-Cre;*Pkd1*^flox/flox^) mice administered DW, 15 mg/kg of HL156A, or 25 mg/kg of HL156A. N/S: not significant. ** *p* < 0.01.

**Figure 4 biomolecules-14-00806-f004:**
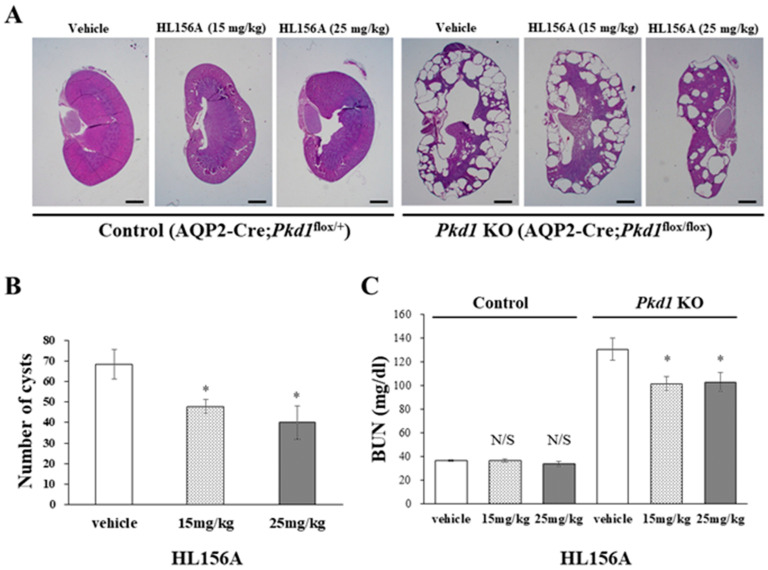
HL156A reduces the number and size of cysts in the collecting duct-specific *Pkd1* knockout (KO) mice. (**A**) Hematoxylin- and eosin-stained kidney sections from control (AQP2-Cre;*Pkd1*^flox/+^) and *Pkd1* KO (AQP2-Cre;*Pkd1*^flox/flox^) mice administered distilled water (DW) as the vehicle or HL156A. Scale bar, 1 mm. (**B**) The number of cysts from *Pkd1* KO (AQP2-Cre;*Pkd1*^flox/flox^) mice administered HL156A (15 mg/kg or 25 mg/kg). (**C**) Blood urea nitrogen (BUN) levels were measured in control (AQP2-Cre;*Pkd1*^flox/+^) and *Pkd1* KO (AQP2-Cre;*Pkd1*^flox/flox^) mice administered DW as the vehicle or HL156A. N/S: not significant. * *p* < 0.05.

**Figure 5 biomolecules-14-00806-f005:**
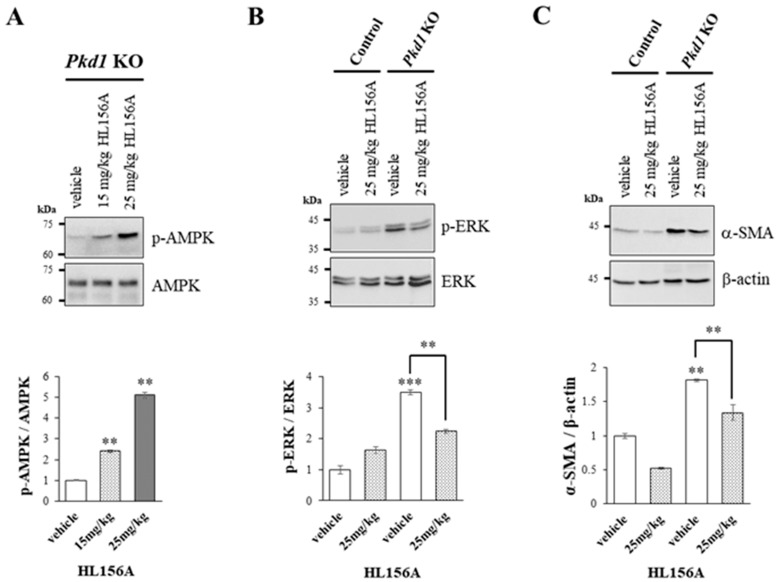
Increased phosphorylation of AMPK by HL156A reduces the phosphorylation of ERK and the expression of SMA. (**A**) Immunoblot of the phosphorylation of AMPK by HL156A in *Pkd1* knockout (KO) (AQP2-Cre;*Pkd1*^flox/flox^) mice. (**B**) Immunoblot of phospho-ERK and ERK from control (AQP2-Cre;*Pkd1*^flox/+^) and *Pkd1* KO (AQP2-Cre;*Pkd1*^flox/flox^) mice administered with distilled water (DW) as the vehicle or 25 mg/kg of HL156A. (**C**) Immunoblot of α-SMA from control (AQP2-Cre;*Pkd1*^flox/+^) and *Pkd1* KO (AQP2-Cre;*Pkd1*^flox/flox^) mice administered distilled water (DW) as the vehicle or 25 mg/kg of HL156A. β-actin was used as the loading control. The intensity of protein bands was quantified using image J. ** *p* < 0.01; *** *p* < 0.001.

## Data Availability

All data are included within the article. Raw data requests and further enquiries can be directed to the corresponding author.

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
