# Peer review of "HL156A, an AMP-Activated Protein Kinase Activator, Inhibits Cyst Growth in Autosomal Dominant Polycystic Kidney Disease"

_biomolecules, 2024, doi:10.3390/biom14070806_

Round 1

Reviewer 1 Report

Comments and Suggestions for Authors

The topic selection is novel and focuses on an in-depth study of treatment for autosomal dominant polycystic kidney disease (ADPKD), a prevalent genetic kidney disorder. As the treatment of ADPKD has always been a clinical challenge, exploring new therapeutic approaches holds significant importance. HL156A, as a new and potent AMPK activator, possesses the potential to inhibit cyst growth in ADPKD, providing a new direction for disease treatment. The research methodology is reasonable, employing a combination of in vitro and in vivo experiments to comprehensively evaluate the inhibitory effect of HL156A on the proliferation of ADPKD cyst cells. In vitro experiments utilized human telomerase reverse transcriptase (hTERT)-immortalized renal cyst cells from ADPKD patients, validating the inhibitory effect of HL156A through an MTT assay. In vivo experiments further confirmed the cyst-inhibitory effect of HL156A by generating a Pkd1 conditional knockout mouse model. This comprehensive research strategy lends credibility to the results.

However, there are some limitations in the article:

1.     The mechanism of action of HL156A has not been thoroughly explored, remaining at a descriptive level. To enhance the depth of the study, I suggest the authors delve deeper into how HL156A inhibits cyst growth through AMPK activation. 

2.     The safety profile of HL156A has not been assessed, which is crucial for practical application. 

3.     Why were only 15 and 25 mg/kg doses selected in the animal model, without conducting a dose-response relationship study? 

4.     Regarding the details, in Figures 2 and 4, the lack of scale bars in the HE-stained and immunofluorescence images makes it difficult to accurately determine the size, distribution, or morphology of cells, as well as their comparison with normal tissue.

Author Response

Comments 1: The mechanism of action of HL156A has not been thoroughly explored, remaining at a descriptive level. To enhance the depth of the study, I suggest the authors delve deeper into how HL156A inhibits cyst growth through AMPK activation.

Response 1: We described “It is reported that activated AMPK phosphorylates BRAF on serine 729 (S729). This phosphorylation disrupts their interaction with the scaffold protein Kinase Suppressor of RAS 1 (KSR1) and promotes the association of BRAF with 14-3-3 proteins, thereby attenuating mitogen-activated protein kinase (MEK)/ERK signaling pathway [32]. Therefore, it is reasonable to conclude that activated AMPK by HL156A, an AMPK activator, phosphorylates BRAF and blunts the BRAF/MEK/ERK signaling pathway, and that the reduction of ERK phosphorylation by HL156A inhibits cyst growth and decreases kidney mass in Pkd1 KO mice.” to explain how HL156A inhibits cyst growth through AMPK activation in discussion part.

Comments 2: The safety profile of HL156A has not been assessed, which is crucial for practical application. 

Response 2: We described “HL156A, also named IM156, demonstrated a good safety profile and tolerability at the recommended Phase 2 clinical dose” in discussion part and added reference assessed the safety profile of HL156A.

Comments 3: Why were only 15 and 25 mg/kg doses selected in the animal model, without conducting a dose-response relationship study?

Response 3: Although we did not mention it in the paper, we conducted animal experiments with various doses. When a dose of 30 mg/kg was injected into mice, they died, so we conducted experiments with doses less than 30 mg/kg. We described “When mice were injected with a dose of 30 mg/kg of HL156A starting from P2, they did not survive, so the experiment was conducted with a dose of 25 mg/kg or less.” in Materials and Methods part.

Comments 4: Regarding the details, in Figures 2 and 4, the lack of scale bars in the HE-stained and immunofluorescence images makes it difficult to accurately determine the size, distribution, or morphology of cells, as well as their comparison with normal tissue.

Response 4: We added the scale bars in the HE-stained and immunofluorescence images of Figure 2 and 4.

Reviewer 2 Report

Comments and Suggestions for Authors

In this study, authors compared a new drug (HL156A), a metformin analogue, with metformin in reducing the growth of kidney cysts in a mouse model (Pkd1 KO)

Rationale

- cAMP is a key messenger in cyst growth and fluid accumulation in ADPKD.

- AMPK activation by metformin seems to reduce cyst growth in mouse models of ADPKD

-This study aims to demonstrate superiority of HL156A a potent AMPK activator, in comparison to metformin, to treat ADPKD in mouse.

 Methodology

-         -  Authors started testing the viability of human renal cyst cells treated with increasing concentrations of HL156A

-          - A collecting duct specific Pkd1 KO mice was created since homozygous Pkd1 mutated mice die in utero

-          -HL156A was administered in a control group of mice (no kidney cysts) and a Pkd1 KO mice for 28 days

-         - Kidney tissue samples were used for western blot analysis and for histology and immunofluorescence analysis.

Results

-         - When compared to metformin, HL156A significantly inhibited cell viability in concentration as low as 5µM.

-         - Pkd1 KO mice showed expected increased kidney volume and morphological evidence of cyst dyscrasia over time.

-        -  Immunofluorescence demonstrated that cysts were located to collecting duct.

-          -HL156A reduces kidney mass and the number and size of cysts in Pkd1 KO mice

-         - Increasing dose of HL156A induces increasing activation of AMPK suggesting a role of this pathway in cyst formation and growth

-          -The expression of SMA, a marker of fibrosis, increases in Pkd1 KO mice  but decreases after HL156A treatment.

Minor issue

- line 49/50 – besides PKD1 and PKD2 other genes were associated with PKD

Author Response

Comments 1: line 49/50 – besides PKD1 and PKD2 other genes were associated with PKD.

Response 1: We described “besides PKD1 and PKD2 other genes were associated with PKD.” in introduction part and added the reference.